# First Immunoassay for Measuring Isoaspartate in Human Serum Albumin

**DOI:** 10.3390/molecules26216709

**Published:** 2021-11-05

**Authors:** Jijing Wang, Susanna L. Lundström, Sven Seelow, Sergey Rodin, Zhaowei Meng, Juan Astorga-Wells, Qinyu Jia, Roman A. Zubarev

**Affiliations:** 1Department of Medical Biophysics and Biochemistry, Karolinska Institute, 171 77 Stockholm, Sweden; jijing.wang@ki.se (J.W.); susanna.lundstrom@ki.se (S.L.L.); sven@lifescienceinnovationconsulting.com (S.S.); sergey.rodin@ki.se (S.R.); zhaowei.meng@ki.se (Z.M.); juan.astorga.wells@pelagobio.com (J.A.-W.); qinyu.jia@ki.se (Q.J.); 2Department of Surgical Sciences, Uppsala University, 752 36 Uppsala, Sweden; 3Endocrinology Research Centre, 115478 Moscow, Russia; 4HDXperts AB, 183 48 Danderyd, Sweden; 5Department of Pharmacological & Technological Chemistry, I.M. Sechenov First Moscow State Medical University, 119435 Moscow, Russia; 6The National Medical Research Center for Endocrinology, 115478 Moscow, Russia

**Keywords:** in vitro diagnostics, blood analysis, monoclonal antibody (mAb), mass spectrometry, enzyme-linked immunosorbent assay (ELISA), human serum albumin (HSA)

## Abstract

Isoaspartate (isoAsp) is a damaging amino acid residue formed in proteins mostly as a result of spontaneous deamidation of asparaginyl residues. An association has been found between isoAsp in human serum albumin (HSA) and Alzheimer’s disease (AD). Here we report on a novel monoclonal antibody (mAb) 1A3 with excellent specificity to isoAsp in the functionally important domain of HSA. Based on 1A3 mAb, an indirect enzyme-linked immunosorbent assay (ELISA) was developed, and the isoAsp occupancy in 100 healthy plasma samples was quantified for the first time, providing the average value of (0.74 ± 0.13)%. These results suggest potential of isoAsp measurements for supplementary AD diagnostics as well as for assessing the freshness of stored donor blood and its suitability for transfusion.

## 1. Introduction

Ammonia (NH_3_) loss from an asparagine (Asn) residue and water loss from an aspartic acid (Asp) residue in proteins result in the unstable cyclic intermediate of succinimide. The latter quickly hydrolyzes (attaches a water molecule) to become a ß-isomer (isoaspartyl—isoAsp) of aspartic acid (Asp) [1]. IsoAsp differs from Asp by the absence of a CH_2_ group in the side chain and its insertion after the backbone alpha-carbon, which not only distorts α-helices and ß-sheets of the protein structure but can also affect protein function [2]. IsoAsp formation occurs spontaneously at physiological conditions, and it speeds up in alkaline solutions or at a higher temperature [3]. Although in most organisms the protein-L-isoaspartyl (D-aspartyl) O-methyltransferase (PIMT) can repair IsoAsp residues, converting them back to Asp in a two-stage process using as a cofactor S-adenosylmethionine (SAM), such repair is only 15–25% efficient [4]. Therefore, isoaspartyl residues can accumulate in long-lived proteins, such as human serum albumin (HSA) with an average lifetime in blood of 28 days. The isoAsp accumulation can result in a loss of native structure, diminishing protein solubility. As the isoaspartyl peptide bonds are also difficult to degrade proteolytically [5,6], the isoAsp-containing proteins are prone to aggregate. The presence of isoAsp may also render the protein immunogenic [7]. As the isoAsp repair mechanism weakens with age, an excess of isoAsp is building up in proteins of aging organisms, which is hypothesized to result in deteriorated health and ultimately neurodegeneration [8,9,10].

The hallmark of Alzheimer’s disease (AD) is the aggregation in brain of the amyloid beta (Aß) peptide [11,12,13,14]. In the early 1990s, it was suggested that in the beginning of the cascade leading to AD could be the isoAsp formation in Aß, particularly in position 7 of its sequence (isoAsp7-Aß) [15,16,17,18]. A set of mAbs against isoAsp7-Aβ have recently been developed, while previously no mAb specific to isoAsp in peptides or proteins had been reported. Experiments on mouse models have indicated that these mAbs have therapeutic potential [19]. Yet, isoaspartyl occurs and accumulates not only in Aß, which is mainly present in brain, but also in all proteins in the organism, including blood proteins. Recently, blood biomarkers of AD have gained much attention, as they allow for earlier diagnostics of the disease [20,21,22,23]. Even though most of these biomarkers relate to Aß and hyperphosphorylated tau protein (another important AD hallmark), other blood proteins have also been strongly implicated in AD, including the most abundant blood protein, HSA [24,25,26]. HSA is the major transporter of Aß in blood (>90%), playing a key role in Aß clearance and inhibiting its fibrillation. As the isoAsp level in HSA may be characteristic of the general state of isoAsp repair in a given organism, and thus can possibly relate to a plethora of pathologies, there is a need to measure this level. Liquid chromatography in combination with tandem mass spectrometry (LC-MS/MS) is a versatile discovery tool with isoAsp quantification capabilities [8], but a simpler and more accessible analytical method is enzyme-linked immunosorbent assay (ELISA). No antibodies specific to isoAsp in HSA are currently available commercially. Thus, we designed for this purpose such a monoclonal antibody (mAb), the first of its kind. In this article, we describe the mAb development and characterization, as well as its application to studying the distribution of isoAsp occupancies in HSA of a normal (healthy) cohort.

The anti-isoAsp antibody development is a nontrivial task given low immunogenicity of isoAsp and the small size of its chemical group. This can explain the absence of pan-isoAsp antibodies. Therefore, we first selected the most characteristic deamidation site in the HSA sequence and then produced murine mAbs against a peptide containing this sequence. Two clones targeting artificially aged HSA (aHSA) with vastly different specificities were identified, and the related genes were sequenced by genomics methods. The difference between these sequences revealed the origin of mAb specificity to isoAsp. Then the most specific mAb, 1A3, was purified and the nucleotide sequence was verified by sequencing the mAb de novo by LC-MS/MS. The presence of posttranslational modifications, such as glycans, was also established. Their effect on mAb specificity was studied by deglycosylating the mAb with five complementary enzymes and measuring the changes in binding to aHSA. We further characterized the kinetics of 1A3 binding to aHSA with surface plasmon resonance (SRP) and verified the sites of mAb-antigen interaction with hydrogen-deuterium exchange mass spectrometry (HDX-MS). We then transferred the antigen-specific region of 1A3 to another IgG scaffold, identifying the optimal for specificity IgG scaffold. Finally, we created and calibrated a quantitative assay for measuring isoAsp in blood HSA and applied it to a cohort of 100 healthy individuals, determining for the first time the normal range of isoAsp in HSA.

## 2. Results

### 2.1. Identification of the “IsoAsp Meter” Peptide in HSA 

Among the four quantified *by* LC-MS/MS Asn-containing tryptic HSA peptides, LVNEVTEFAK exhibited the highest deamidation rate in the course of HSA incubation, reaching the occupancy of ≈60% at day 28 (Figure 1b). This peptide is exposed to the solvent in the native protein structure (Figure 1c), which explains its high isoAsp formation rate, and makes it easily available for binding with antibodies. Moreover, this peptide is located in domain I of HSA that exhibits stronger binding affinity to Aß monomer compared to other domains and disrupts the structural transformation of Aß42 protofibrils to fibrils [27]. The deamidation rates of other Asn-containing peptides were proportional to the isoAsp occupancy in LVNEVTEFAK. Thus, this peptide was selected as an “isoAsp meter” reflecting the deamidation state of asparaginyls in the whole HSA molecule.

### 2.2. Comparison Results of Different ELISAs

From comparison of different ELISA methods (Table 1), we found that the indirect ELISA with chemiluminescence detection has the highest sensitivity with the signal to noise ratio (S/N) of 39.9 at 6% isoAsp and 4.3 at 0.6% isoAsp. Moreover, the cost including the plate, antibodies and solutions was the lowest among all ELISAs. Thus, the indirect ELISA was selected for further isoAsp analyses.

### 2.3. Indirect ELISA of Deamidated HSA

By calculating the ratio of the chemiluminescence signal produced in indirect ELISA by aHSA versus fresh HSA (fHSA), the specificity of the 1A2 and 1A3 mAbs was determined, with 1A3 showing overwhelmingly higher specificity (Figure 1d). A criss-cross analysis showed that the maximum specificity is obtained when the concentration of primary 1A3 antibody is 800 ng/mL and the dilution ratio of the secondary antibody is 1:10,000 (Appendix A). No cross-reactivity with deamidated human transferrin (also abundant in human blood) as well as myoglobin from equine skeletal muscle was detected (Appendix A).

### 2.4. Sequence Analysis of mAb Variable Region

The nucleotide sequences obtained by two complementary DNA analyses of 1A2 and 1A3 were identical for each mAb and the corresponding amino acid sequence of 1A3 was 100% confirmed by two independent de novo amino acid sequencing efforts (Figure 2a–d). Only one amino acid in each of the light and heavy chains turned out to be different between the specific 1A3 and unspecific 1A2 clones: replacement of Ile-51 by Ser in the heavy chain and Leu-60 by Ser in the light chain decreased the specificity to isoAsp by three orders of magnitude. It was also confirmed that the immunoglobulin (Ig) isotype of 1A2 was of the IgM type and 1A3–of the IgG3 type.

We additionally verified the above results by LC-MS/MS of 1A3 mAb tryptic digest, obtaining 100% sequence coverage and confirming Ile in position 51 of V_H_ and Leu in position 60 of V_L_. The de novo sequencing analyses also found a N-glycosylation at Asp-302 in the constant region of heavy chain (C_H_), with 16 different glycans attached. Additional O-glycosylation was detected on C_H_ between the positions 227 and 244. Perhaps more importantly, glycosylation in the variable region V_L_ was found of both N-linked (positions 28 and 30) as well as O-linked (positions 22, 26 and 34) types (Figure 2e, Table 2). The impact of these glycosylations on the specificity of 1A3 was however found to be limited (see below). 

### 2.5. Kinetic Analyses

The SPR-determined kinetics binding parameters of 1A3 with aHSA are listed in Table 3. The 1A3 sensorgrams demonstrates a clear-cut concentration-response relationship with smooth and consistent curves indicative of good interactions (Figure 3a). The calculated K_D_ value of 2.1 × 10^−8^ M means an above-average (≈10^−6^ M [28]) binding affinity.

### 2.6. Paratope Mapping

After optimization, 51 reporter peptic products covering 86% of the 1A3 sequence were chosen for deuterium uptake kinetics due to their high identification scores, good signal-to-noise ratios, and the absence of an overlap with other molecules in the mass spectra. The background deuterium incorporation without the antigen is shown in Figure 3b. The differential map on Figure 3c reflecting the changes in deuterium incorporation when the antigen was added reveals the region “FIRNKANGYTTE” (residues 50–61 in VH) as exhibiting the highest reduction in deuterium incorporation, consistent with shielding of this region by the bound antigen. This result confirmed Ile-51 to be critical for 1A3 specificity (see above), while the region containing the other specificity-implicated amino acid, Leu-60 from VL, showed in HDX-MS no significant difference in the presence of antigen. The other region highlighted by HDX-MS was DHY (residues 31–33); however, this may be due to an allosteric structure change in mAb.

### 2.7. Glycosylation Effect on 1A3 Specificity

To reveal the impact of glycosylation on 1A3 specificity, the mAb was deglycosylated by five different enzymes. The specificity diminished most (−16%) after deglycosylation by ß1,4-galactosidase. Deglycosylation by N-glycosidase F, α2-3,6,8,9-neuraminidase and endo-α-N-acetylgalactosaminidase weakened somewhat the 1A3 specificity as well, while ß-N-acetylglucosaminidase did not produce significant changes (Figure 4a). Overall, glycosylation does not seem to affect the 1A3 specificity dramatically.

### 2.8. Effect of the Ig isotype

The sequences of VL and VH of 1A3 were incorporated into the IgG2B scaffold and the specificity of thus obtained 1A3* mAb to aHSA versus fHSA was tested. Figure 4b demonstrates that, while the specificity was still greatly exceeding that of 1A2, it was significantly lower than that of 1A3.

### 2.9. Determination of IsoAsp Levels in Healthy Plasma

We used the developed indirect ELISA to quantify the isoAsp levels in plasma samples from 100 healthy donors (59 males and 41 females). A linear or second-order polynomial standard curve linking the chemiluminescence intensities with the known isoAsp levels in a set of standard mixtures of aHSA and fHSA provided accurate calibration. At 0.10% isoAsp content, the typical signal to noise (S/N) ratio was 3.1, where the ELISA signal from fresh HSA was taken as the noise level. The results fit normal distribution with the average isoAsp level of 0.74% using linear calibration (Figure 4c) and 0.96% for a second-order polynomial calibration, with standard deviations of 0.13% and 0.14%, respectively. Note that both average values overlap within their standard errors. The results repeated with a two-week interval correlated with R > 0.95 (Figure 4d). The average isoAsp occupancy in female blood was 0.3% higher than in the male samples. A slight anti-correlation between the isoAsp occupancy and donor age was also noticed.

## 3. Discussion

In the current study, a novel 1A3 mAb against an isoAsp site in HSA, the first of its kind, was raised and thoroughly characterized. The ELISA analyses based on 1A3 showed that this mAb is highly specific to the HSA sequence LVNEVTEFAK, where N is converted to isoAsp, while having negligible affinity to unmodified HSA. Importantly, 1A3 is not cross-reactive with other deamidated blood proteins, which ensures the specificity of HSA analysis in such a complex matrix as blood plasma. The lowest isoAsp occupancy in HSA of healthy human blood was found to be 0.4%, and as at 0.1% isoAsp the S/N ratio already exceeded 3, the accuracy of measurements was not limited by the assay sensitivity. The higher average isoAsp level in female blood was consistent with our earlier findings [8] as well as in general lower levels of isoAsp repair [29,30] and immune response [27] in females. On the other hand, the found slight negative correlation with age requires further investigation. In principle, isoAsp repair and anti-isoAsp defenses (proteases, antibodies) become less active with age, which would logically result in isoAsp accumulation in elderly individuals. However, such accumulation would likely cause a disease in such individuals [8], which would have precluded their inclusion in the healthy cohort.

As elevated isoAsp levels in blood are associated with AD [8], an ELISA with 1A3 mAb could be used as an auxiliary tool in AD diagnostics. The fact that the targeted isoAsp residue resides in the HSA domain responsible for Aß binding and disrupting the structural transformation of Aß42 protofibrils to fibrils [28], potentially links deamidation at this site this to disease etiology. Although significant amount of preclinical and clinical research needs to be done for the assay validation, adding this analysis to already formidable arsenal of AD diagnostics tools will not incur much inconvenience and additional expense. This is because of the simplicity and low cost of indirect ELISAs compared to cerebrospinal fluid (CSF) testing, magnetic resonance imaging (MRI) and especially positron emission tomography (PET) imaging that are conventionally used in AD diagnostics.

The developed mAb can also be used in analysis of blood stored in biobanks. Blood is usually separated into red blood cells and plasma and is stored for up to 12 months (standard storage time), but sometimes for even 24 to 36 months. While deamidation rate slows down with temperature, it reaches zero at ≈−70 °C (Zubarev, R.A. et al., 2011, unpublished data), and thus deamidation still occurs during the storage at −18 to −24 °C often used in blood banks. Globally, 28 million liters of blood plasma are separated annually from donor blood, and about one million liters are discarded due to the expiration of shelf life [31]. Whereas there are certain assays for measuring blood coagulation properties, isoAsp levels are not monitored. Therefore, there is potential for saving at least a fraction of that blood that still has acceptable isoAsp levels. The opposite is also true–some fresh blood, even from not very old donors, may contain high isoAsp levels, which makes it unsuitable for transfusion.

## 4. Materials and Methods

### 4.1. Artificial Deamidation of Proteins

We artificially deamidated HSA by incubating the protein in Tris buffer at pH 8.5 and 60 °C for 0, 7, 14, and 28 days. After digesting the artificially aged and totally fresh HSA (aHSA and fHSA, respectively) by a mix of the enzymes Lys-C and trypsin, we desalted the peptides and analyzed them by LC-MS/MS with electron transfer dissociation (ETD), which provides an isoAsp-specific fragment [32]. Then we identified the isoAsp-containing peptides, quantified the peptide abundances by our home-made software Quanti [33] and calculated the isoAsp occupancy in the Asn-containing peptides as a function of ageing duration (Figure 1a).

For control, human holo-transferrin and myoglobin from equine skeletal muscle were aged for 7 days following the same protocol as described above.

### 4.2. Generation of mAbs against the “IsoAsp Meter” in HSA

After the peptide with the highest rate of isoAsp occupancy increase during ageing process was chosen, its variants with Asn, Asp and isoAsp were synthesized. Hybridoma clones expressing murine mAbs specific to the isoAsp peptide but not to Asn or Asp peptide were then produced. The clones 1A2 and 1A3 showing the highest specificity were selected for further investigation. In order to find the best method for isoAsp level quantification, we tested the specificity and sensitivity of indirect ELISA (via primary and secondary antibodies) and sandwich ELISA (via capture, detection and secondary antibodies) using chemiluminescence and absorbance for detection, and chose indirect ELISA as the best method showing the highest signal to noise (S/N) ratio.

### 4.3. Indirect ELISA

The antigens aHSA (≈60% isoAsp) and fHSA (~0% isoAsp) were diluted to 10 µg/mL by Phosphate Buffered Saline (PBS) (Cytiva, Marlborough, MA, USA), pH 7.4. The 96-well white opaque polystyrene plates were coated with 50 µL/well of the aHSA and fHSA diluted in PBS and incubated at RT for 2 h. Plates were washed in 300 µL/well PBS containing 0.05% Tween-20 (Sigma-Aldrich, St. Louis, MO, USA) (PBST) three times and shaken dry. The vacant sites were blocked with 200 µL/well 10% skim milk powder (Sigma-Aldrich, St. Louis, MO, USA) in PBST at 4 °C for overnight and washed with PBST as above. Then 50 µL/well of mAb as primary antibody in blocking buffer was added and incubated at RT for 2 h. After washing with PBST, a goat anti-mouse IgG Conjugated to Horseradish Peroxidase (Jackson Immuno Research, Cambridge, UK) was used as secondary antibody and incubated at RT for 2 h. Working solution (100 µL/well) was prepared according to the SuperSignal ELISA Pico Chemiluminescent Substrate protocol (Thermo Scientific, Waltham, MA, USA), and chemiluminescence intensities were measured immediately by the luminometer Tecan Infinite M200 Pro (Tecan, Männedorf, Switzerland).

### 4.4. DNA Sequence Analysis of mAbs Variable Regions

The DNA sequences of both 1A2 and 1A3 mAbs were analyzed independently by two companies, Fusion Antibodies (Belfast, Northern Ireland) and Absolute Antibody (Oxford, UK). The analyses done by Fusion Antibodies include the following procedures. The mRNA was extracted from the hybridoma cell pellets, and RNA was extracted from the pellets using the RNeasy Plus Mini Kit (Qiagen, Hilden, Germany). A cDNA library was created from the extracted RNA by reverse-transcription with an oligo (dT) primer and used as a template to amplify the variable (V) regions of both H and L chains of the mAbs by PCR reactions. The V_H_ and V_L_ products were cloned into the Invitrogen sequencing vector pCR2.1, transformed into TOP10 cells and screened by PCR for positive transformants. Selected colonies were picked and analyzed by DNA sequencing on an 3130xl Genetic Analyzer (Applied Biosystems, Waltham, MA, USA).

The Absolute Antibody used a proprietary sequencing approach, whole transcriptome shotgun sequencing (RNA-Seq). Total RNA was extracted from cells and a barcoded cDNA library was generated through RT-PCR using a random hexamer. Next Generation Sequencing was performed on a HiSeq sequencer (Illumina, San Diego, CA, USA). Contigs were assembled using a proprietary approach and data mined for antibody sequences identifying all viable antibody sequences (i.e., those not containing stop codons). Variable heavy and variable light domains were identified separately, and the relative abundance of each identified gene was reported in transcripts per million (TPM). The species and isotype of the identified antibody genes were confirmed. Sequences were compared with known aberrant (i.e., non-functional) antibody genes that are present in many hybridomas and these genes were removed from analysis when necessary.

### 4.5. De Novo Protein Sequencing of 1A3

Verification of the amino acid sequences of 1A3 mAb was performed independently by Peaks AB (Ontario, CA) and Rapid Novor Inc (Ontario, CA). The Antibody Characterization Service of Peaks AB performed in-solution endoproteinase digestions of 1A3. In brief, after reduction by dithiothreitol and alkylation by iodoacetamide, the sample was equally divided into 3 aliquots for three kinds of enzyme digestions: (a) Lys-C and trypsin, (b) Lys-C and Glu-C, (c) Glu-C (Sigma-Aldrich, St. Louis, MO, USA) only, following manufacturer’s instructions. The digests were desalted and analyzed on the LC-MS/MS. Both full MS scans and MS2 scans were acquired in the high resolution Orbitrap mass analyzer. MS2 data were acquired using HCD and ETD followed by HCD (EThcD) fragmentation methods. The raw data files were processed using the PEAKS Studio software. The Leu/Ile differentiation was performed using the EThcD MS/MS data and PEAKS AB I/L differentiation algorithm, with subsequent manual inspection.

The amino acid sequencing method by Novor was similar as Peaks, with eleven digestions prepared using six different enzymes (Nonspecific, Pepsin, Trypsin, Chymotrypsin, Asp N, Lys-C) (Sigma-Aldrich, St. Louis, MO, USA). All samples were analyzed by LC-MS/MS using a Fisher Orbitrap Fusion mass spectrometer (Thermo Fisher Scientific, Waltham, MA, USA). The isobaric Leu and Ile residues were determined based on (a) w-ion in the data when the WILD™ option was included; (b) enzyme specificity at I and L in chymotrypsin and pepsin digestions; (c) statistical residue distribution and homologous sequences in the antibody database.

### 4.6. Surface Plasmon Resonance (SPR)

Kinetics of the interaction between 1A3 mAb and aHSA were measured by ProteoGenix (Schiltigheim, France) using the Biacore T2000 SPR biosensor (Cytiva, Marlborough, MA, US). After immobilization of 10 μg/mL aHSA (antigen) in 20 mM sodium acetate on a CM5 sensor chip, the mAb was serially diluted starting from 128 nM concentration in a buffer containing 10 mM HEPES (pH 7.4), 150 mM NaCl, 3 mM EDTA and 0.01% Surfactant P20. Then the buffer was flown over the CM5 chip and responses were captured over an association/dissociation cycle. After the diminishing concentrations were successively tested, the kinetics parameters *k*_d_ (dissociation rate constant) and *k*_a_ (association rate constant) were calculated using the BIAevaluation software. The dissociation constant *K*_D_ was calculated as *K*_D_ = *k*_d_/*k*_a_.

### 4.7. Hydrogen-Deuterium Exchange Mass Spectrometry (HDX-MS)

The 1 mg/mL solution of 1A3 mAb in PBS was mixed with 1 mg/mL aHSA solution in PBS in a 1:1 molar ratio, incubated for 2 h at RT. The formed Ag/mAb complex was washed with PBS three times and concentrated to 40 µL using a Amicon Ultra-0.5 mL Centrifugal filter (10 K) units (Merck, Darmstadt, Germany). The control sample was prepared via the same procedure, but without adding aHSA. The automated HDX-MS system (Biomotif HDX CTC PAL, Sweden) was used for automated labelling, quenching, digestion, cleanup and separation of samples at 2 °C. In details, after labelling with deuterated PBS, the reaction was quenched by adding a solution containing 4 M Urea, 100 mM tris(2-carboxyethyl)phosphine (TCEP) and 0.05% trifluoroacetic acid. Samples were digested on an immobilized pepsin column (2.1 × 30 mm) for 2 min at a flow rate of 60 μL/min, followed by an on-line desalting step using a 2 mm I.D. × 10 mm length C-18 pre-column (ACE HPLC Columns, Aberdeen, UK) using 0.1% formic acid at 400 μL/min for 1 min. Peptic peptides were then separated by a 18 min 8–55% linear gradient of acetonitrile in 0.1% formic acid using a 2 mm I.D. × 50 mm length HALO C18/1.8 μm analytical column operated at 60 μL/min. Then a Q-Exactive Orbitrap mass spectrometer (Thermo Fisher Scientific, Waltham, MA, USA) was operated at mass resolution 140,000 at m/z 400 in all experiments at RT. All HDX-MS data was processed by the software HDExaminer (Sierra Analytics, Version 2.5.1). Peptides were identified by Mascot search engine (Matrix Science, Version 2.5.1) using a dedicated database containing the sequences of 1A3 variable region, using the 10 ppm precursor mass tolerance for MS and 0.05 Da fragment mass tolerance for MS/MS data.

### 4.8. Deglycosylation of 1A3 mAb

To test if the specificity of 1A3 mAb changes after deglycosylation, we used five different enzymes from the glycoprotein deglycosylation kit (Sigma-Aldrich, Taufkirchen, Germany) according to the manufacturer’s protocol. In brief, 10 µL 5× Reaction Buffer was added to 5 tubes of 3 µg/µL 1A3 mAb in deionized water (35 µL/tube), followed by 1 µL of N-Glycosidase F, α2-3,6,8,9-Neuraminidase, ß-N-Acetylglucosaminidase, ß1,4-galactosidase and endo-α-N-acetylgalactosaminidase respectively and 4 µL of deionized water. For a control, another tube containing only the same amount of 1A3 mAb and 5× Reaction buffer topped with water was prepared. After incubation at 37 for 48 h, all samples were diluted to 800 ng/mL in blocking buffer as above for an indirect ELISA analysis on fHSA and aHSA.

### 4.9. Production of 1A3* mAb

Absolute Antibody (Oxford, UK) produced 1A3* mAb with the same variable region sequences as 1A3, but of a different IgG isotype (IgG2b).

### 4.10. ELISA of Blood Plasma from Healthy Donors

Plasma from a 100-strong cohort of healthy blood donors (Appendix A) was obtained from ProMedDx Limited Liability Company (Norton, MA, USA). The samples were collected under a clinical study that had been reviewed by an Institutional/Independent Review Board (IRB) and/or independent Ethics Committee (IEC) according to the local regulations. On each 96-well plate, 50 donor samples were analyzed with indirect ELISA described above together with calibration samples containing 0%, 0.10%, 0.30%, 0.60%, 1.20% and 2.40% isoAsp obtained by mixing fSHA (presumed to have 0% isoAsp) with aHSA (60% isoAsp). For each 96-well plate independently, linear or a second-order polynomial calibration curve was then fitted to the calibration datapoints, and the data on 50 donor samples were converted to % isoAsp. The analysis was performed in 3 replicates (six 96-well plates altogether).

## 5. Conclusions

Here we developed and characterized 1A3, the first mAb recognizing isoAsp in a structurally and functionally important HSA domain with high specificity and sensitivity as well as with an above-average binding affinity. Using this mAb, we developed a simple and robust ELISA and measured the isoAsp level in 100 healthy plasma samples, determining the normal range of isoAsp occupancy in an important position of HSA for the first time. The provided data opens a way for investigating the potential of using such an ELISA to supplement AD diagnostics, as well as for measuring the freshness of donor blood stored in biobanks and its suitability for transfusion.

## 6. Patents

The Patent Cooperation Treaty (PCT) application is under review.

## Figures and Tables

**Figure 1 molecules-26-06709-f001:**
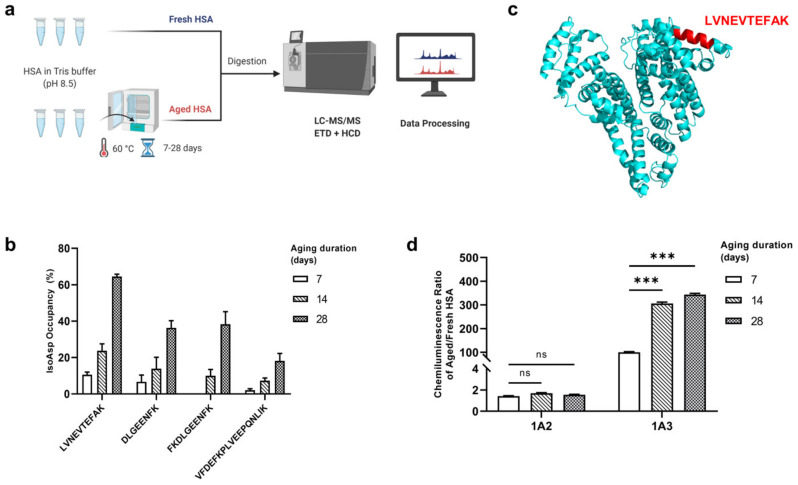
Development of ELISA against isoAsp in human serum albumin (HSA). (**a**) Finding the “IsoAsp meter” peptide for measuring isoAsp in HSA. Artificially deamidated HSA was digested, with the peptides analyzed by LC-MS/MS with ETD and HCD fragmentation and label-free peptide quantification. (**b**) Changes of the isoAsp occupancy in Asn-containing HSA peptides over aging duration. (**c**) Location of the peptide LVNEVTEFAK in the structure of HSA monomer. (**d**) Comparison of the indirect ELISA signals with 1A2 and 1A3 mAbs of HSA aged for 7–14–28 days. Aged HSA was used as a positive control, and fresh HSA as a negative control, while the ratio of these two ELISA signals provided the dynamic range related to mAb specificity, *** *p* < 0.0001, ns: not significant.

**Figure 2 molecules-26-06709-f002:**
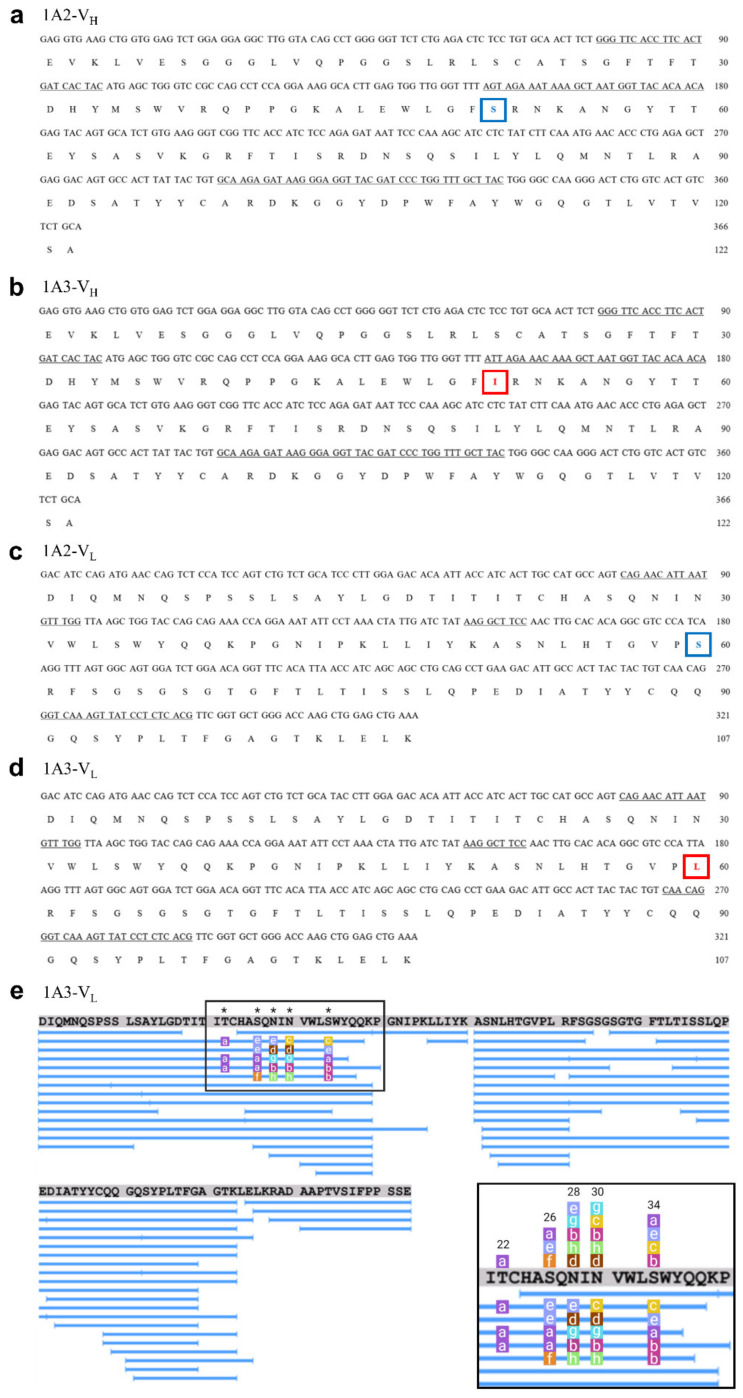
DNA and amino acid sequences of the mAb variable regions. The amino acids differentiating the unspecific 1A2 and specific 1A3 mAbs are highlighted. (**a**) V_H_ of 1A2, (**b**) V_H_ of 1A3, (**c**) V_L_ of 1A2, (**d**) V_L_ of 1A3. (**e**) The coverage of 1A3 variable regions by peptides detected in LC-MS/MS analysis, and the position and type of the detected glycans. Details of attached glycans are given in the inset at the bottom right.

**Figure 3 molecules-26-06709-f003:**
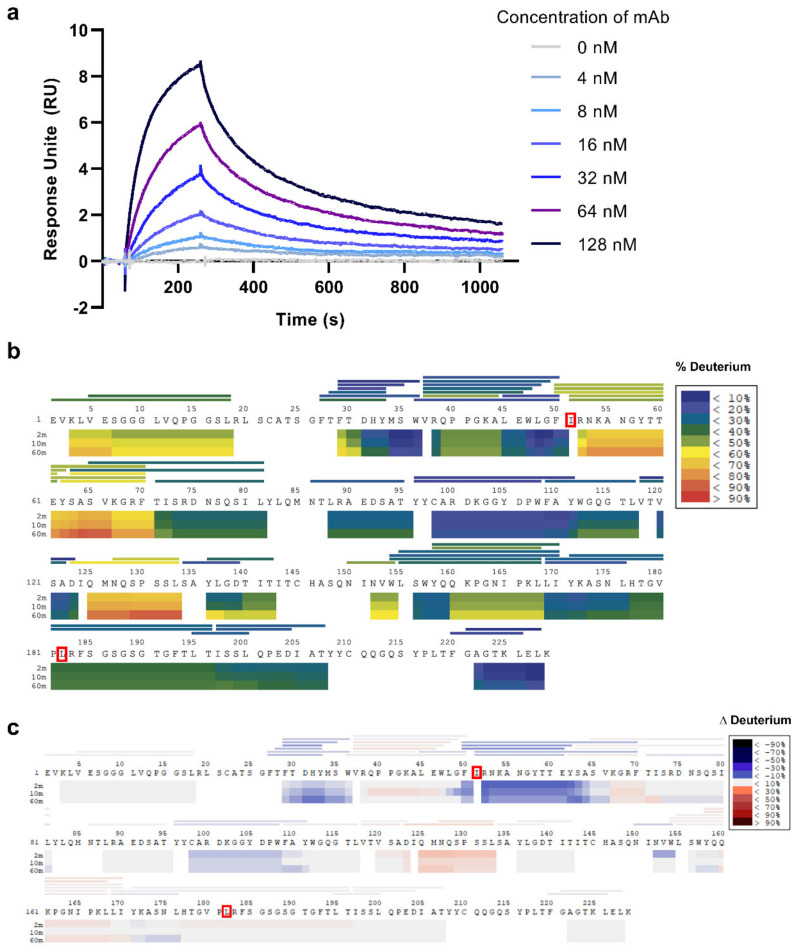
Characterization of binding interface between 1A3 mAb and aHSA. (**a**) The sensorgram of mAb/aHSA interaction at different mAb concentrations. (**b**) The coverage of the 1A3 sequence by reporter peptic fragments (horizontal lines) in HDX MS analysis and the background deuterium incorporation (horizontal bars). (**c**) The differential deuterium incorporation between the mAb/aHSA complex and the mAb alone. Negative values (in blue) represent protection/stabilization while positive values (in red) indicate destabilization.

**Figure 4 molecules-26-06709-f004:**
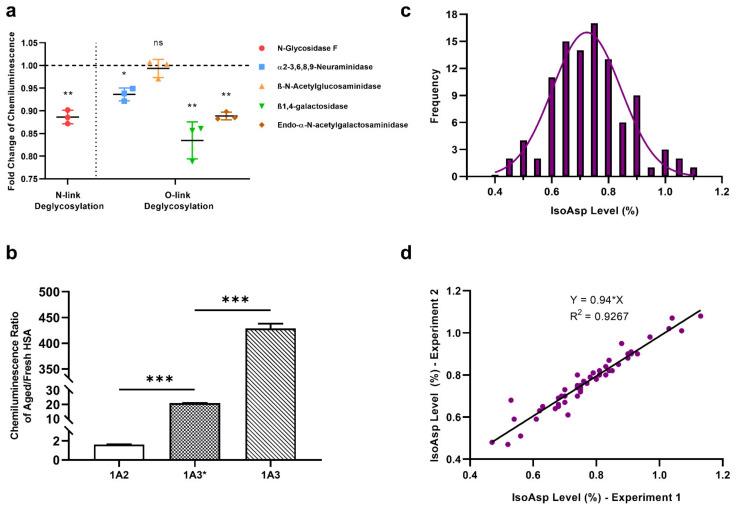
The application of 1A3 mAb in indirect ELISA for determination of isoAsp in blood HSA. (**a**) The changes in 1A3 specificity to aHSA after deglycosylation by five different galactosidases; * *p* < 0.05; ** *p* < 0.01; ns: not significant. (**b**) The comparison of mAbs specificity to aHSA versus fHSA for 1A2, 1A3* and 1A3 mAb, where 1A3* has the same variable region sequences as 1A3 but of a different IgG isotype (IgG2b compared to IgG3); *** *p* < 0.0001 (**c**) The distribution of isoAsp occupancies in HSA measured in 100 blood plasma samples from healthy individuals, obtained using linear calibration. (**d**) Comparison between the ELISA results of 50 blood plasma samples measured using 1A3 antibody with a two-week interval.

**Table 1 molecules-26-06709-t001:** Kinetic binding parameters of the interaction of 1A3 with aHSA.

	Indirect ELISA		Indirect Sandwich ELISA		
Plate	White Polystyrene	Copper Coated	White Polystyrene	White Polystyrene	Copper-coated
Blocking buffer	10% Milk in PBST				
Capture antibody	N/A		15C7 anti-HSA mAb	1A3 mAb	1A3 mAb
Primary/Detection antibody	1A3 mAb		1A3 mAb	1A2 mAb	1E1 anti-HSA mAb
Secondary antibody	Peroxidase AffiniPure Polyclonal Goat Anti-Mouse IgG + IgM (H + L)				
Detection	Chemiluminescence	Absorbance	Chemiluminescence	Chemiluminescence	Absorbance
S/N at 6% isoAsp	39.9	12.1	0.8	0.6	0.8
*p* value	<10^−5^	<10^−5^	7.0 × 10^−3^	<10^−5^	<10^−5^
S/N at 0.6% isoAsp	4.3	3.6	N/A		

**Table 2 molecules-26-06709-t002:** Position, type and mass of glycans in the V_L_ region of 1A3 mAb.

No.	Poition of Glycans	∆M (kD)
a	HexNAc[4]Hex[4]	+1.461
b	HexNAc[4]Hex[3]	+1.298
c	dHex[1]HexNAc[3]Hex[3]	+1.241
d	dHex[1]HexNAc[4]Hex[6]	+1.931
e	dHex[1]HexNAc[4]Hex[4]	+1.607
f	dHex[1]HexNAc[4]Hex[3]	+1.445
g	HexNAc[2]Hex[5]	+1.216
h	HexNAc[4]Hex[5]NeuAc[1]	+1.914

**Table 3 molecules-26-06709-t003:** Kinetic binding parameters of the interaction of 1A3 with aHSA.

Antibody-Antigen	*k_a_* (M^−1^ s^−1^)	*k_d_* (s^−1^)	*K*_D_ (M)
1A3-aHSA	2.6 ± 0.5 × 10^−5^	5.3 ± 0.4 × 10^−3^	2.1 ± 0.3 × 10^−8^

## Data Availability

The mass spectrometry proteomics data have been deposited to the ProteomeXchange Consortium via the PRIDE 34 partner repository with the dataset identifier PXD027563.

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
