# Peer review of "First Immunoassay for Measuring Isoaspartate in Human Serum Albumin"

_molecules, 2021, doi:10.3390/molecules26216709_

Round 1

Reviewer 1 Report

In this paper Wang and colleagues describes their study on Measuring Isoaspartate in Human Serum Albumin by using a new monoclonal antibody that they developed. They focus on the Human Serum Albumin as it has been implicated in Alzheimer’s disease (AD), in addition other biomarkers related to amyloid beta.

The work covers three areas.

  1. The mAb development
  2. Detailed characterization the mAb
  3. Its application to studying g the distribution of isoAsp occupancies in HSA of a normal (healthy) cohort.

Overall the work is good and novel and the study addresses a specific need about lack of pan-isoAsp antibodies due to low immunogenicity of isoAsp and the small size of its chemical group.

Corrections:

Line 41

an excess of isoAsp is building up in proteins

an excess of isoAsp builds up in proteins

line 48

in prior no mAb specific to isoAsp in peptides or proteins had been reported

previously no mAb specific to isoAsp in peptides or proteins had been reported

Line 71

Two most specific clones to artificially aged HSA were identified, and the related genes were sequenced by genomics methods.

It gets confused when the term ‘most specific’ is used to refer to both mabs then later it changes to only one of them being better.

Line 88

aHSA

have not yet defined the abbreviated form so one doesn’t know what aHSA is.

Fig 1d

May be for the reader to get an idea of why 1A2 was specific as well, you need to outline the results of the other Mabs, even if it is in the supp information. Otherwise, as it is one wonders why IA2 is even being considered specific at all.

Also, there is no negative control using an irrelevant antibody. That would help show if 1A2 is specific.

Line 106;

What is S/N? this gets defined later in section 2.9, it should be at the point it is first used.

Line 106.

Is S/N of 12 at 6% correct for indirect ELISA detected by chemiluminescence? Check Table 1. If it is not correct, has the data been correctly interpreted?

Line 133

Define aHSA and fHSA

Line 123

This follows what I commented for Line 71 above, it is either both are specific or not. I think low affinity vs high affinity should be used to avoid this confusion, that is if both are specific.

Line 172

while the specificity was still greatly exceeding that of 1A2, it was an order of magnitude below that of 1A3.

while the specificity was still greatly exceeding that of 1A2, it was significantly lower than that of 1A3.

Fig 2

Legend mixes lower case and upper a, b, c, d, E. which gets confused when gylcans are again lists using alphabets. May be the glycan information should be summarised in a table format instead of adding it all on the figure legend. There is more information in that legend text for a full table than there is in Table 2 that can be described effectively in one sentence.

Line 209

On the other hand, the found slight anticorrelation with age requires more involved explanation.

On the other hand, the observed slight negative correlation with age requires further investigation.

Line 247

home-made software

in-house software

Describe this software or say how it can be accessed otherwise how would other people replicate the results.

Reviewer 2 Report

In the current manuscript by Wang J et al, authors have successfully established the immunoassay to measure isoAsp in human serum albumin and it looks convincing. Authors have performed a good set of experiments to support their data. Overall, it is a well drafted manuscript, has detailed material and methods section, with a good discussion, and enough relevant cited references. I also did not find any similar/parallel literature regarding the claims made in the paper. However, I have few suggestions which I think will improve the quality of the manuscript and will be of additional value for broad scientific community.

Major comments:

There are positive/negative controls missing in Figure 1.

Should include this data:

Authors did criss-cross analysis to conclude their results, however, it will be better if Authors include this data in supplement. Authors claim to show that the “maximum specificity is obtained when the concentration of primary 1A3 antibody is 800 ng/mL” does not have back up with the data. Similar comment to following line “no cross-reactivity with deamidated human transferrin, also abundant in human blood, was detected, as well as myoglobin from equine skeletal muscle”.

Authors have discussed association of isoAsp in human serum albumin and Alzheimer’s Disease in Abstract, Introduction, and Discussion sections however, the usefulness of this novel antibody and ELISA method to detect isoAsp levels will be appreciated if they could already include this data. The paper will benefit largely if authors could show the isoAsp levels between healthy individuals and AD patients to support the usefulness of this novel antibody.

Additional experiment to consider could be checking levels of isoAsp in early vs late Alzheimer’s patients.

I would also encourage to include the data provided in non-published file supplementary file to improve the content of manuscript and discussion.

Minor point:

At some places, improvement in English is needed. Some lines are hard to follow, and it will be great if a language check can be done.

e.g. Introduction, Paragraph line 1 sentence re-arrangement is needed.
